# Fc-Effector-Independent in vivo Activity of a Potent Influenza B Neuraminidase Broadly Neutralizing Antibody

**DOI:** 10.3390/v15071540

**Published:** 2023-07-13

**Authors:** Ahmed M. Khalil, Michael S. Piepenbrink, Ian Markham, Madhubanti Basu, Luis Martinez-Sobrido, James J. Kobie

**Affiliations:** 1Texas Biomedical Research Institute, San Antonio, TX 78245, USA; akhalil@txbiomed.org; 2Department of Zoonotic Diseases, Faculty of Veterinary Medicine, Zagazig University, Zagazig 44511, Egypt; 3Division of Infectious Diseases, University of Alabama at Birmingham, Birmingham, AL 35294, USA; mpiepenbrink@uabmc.edu (M.S.P.); imarkham@uab.edu (I.M.); mbasu@uabmc.edu (M.B.)

**Keywords:** influenza, neuraminidase, antibody, monoclonal, Fc

## Abstract

Influenza B virus (IBV) contributes to substantial influenza-mediated morbidity and mortality, particularly among children. Similar to influenza A viruses (IAV), the hemagglutinin (HA) and neuraminidase (NA) of IBV undergo antigenic drift, necessitating regular reformulation of seasonal influenza vaccines. NA inhibitors, such as oseltamivir, have reduced activity and clinical efficacy against IBV, while M2 channel inhibitors are only effective against IAV, highlighting the need for improved vaccine and therapeutics for the treatment of seasonal IBV infections. We have previously described a potent human monoclonal antibody (hMAb), 1092D4, that is specific for IBV NA and neutralizes a broad range of IBVs. The anti-viral activity of MAbs can include direct mechanisms such as through neutralization and/or Fc-mediated effector functions that are dependent on accessory cells expressing Fc receptors and that could be impacted by potential host-dependent variability. To discern if the in vivo efficacy of 1092D4 was dependent on Fc-effector function, 1092D4 hMAb with reduced ability to bind to Fc receptors (1092D4–LALAPG) was generated and tested. 1092D4–LALAPG had comparable in vitro binding, neutralization, and inhibition of NA activity to 1092D4. 1092D4–LALAPG was effective at protecting against a lethal challenge of IBV in mice. These results suggest that hMAb 1092D4 in vivo activity is minimally dependent on Fc-effector functions, a characteristic that may extend to other hMAbs that have potent NA inhibition activity.

## 1. Introduction

Seasonal influenza vaccines significantly reduce influenza associated morbidity and mortality, but continuing antigenic drift of circulating influenza strains results in substantial variations in efficacy. In the United States, influenza B virus (IBV) infections comprised, on average, ~25% of influenza infections in the 2010–2011 through 2019–2020 seasons and ~1% of infections in the 2021–2022 through 2022–2023 seasons [1]. This reduction is perhaps influenced by the coronavirus disease 2019 (COVID-19) pandemic, the lack of an animal reservoir for IBV, and the recent severe reduction in IBV Yamagata lineage infections [2]. While incidence and severity of IBV infections can vary, children are often disproportionally impacted with IBV infections [3,4,5] and have been overrepresented in many years in pediatric influenza-related deaths [5,6,7]. 

Available treatments for IBV infection include the neuraminidase (NA) inhibitor oseltamivir; however, much higher IC_50_ values have been reported for IBV compared to influenza A viruses (IAV) [8,9]. M2 channel inhibitors are effective against IAV but not against IBV [10,11]. Although not currently approved for use in individuals younger than 5 years old, there are conflicting reports as to whether the cap-dependent endonuclease baloxavir marboxil has comparable activity against IBV to IAV [8,12]. A long-term concern for antiviral efficacy is the potential for increasing the incidence of resistance mutants, which have been described for IBV [13,14,15]. Given the dependence of the efficacy of the seasonal influenza vaccine on adequately matching the predicted circulating strains, and the limitations of current antiviral therapies, new vaccine and therapeutic strategies for the treatment of IBV infections are urgently needed. NA, the second most abundant glycoprotein on the surface of the influenza virus, has gained attention as a therapeutic target in recent years, in part due to its more limited antigenic drift compared to hemagglutinin (HA) and its highly conserved enzymatic site, highlighting its potential for universal vaccine and therapeutic human monoclonal antibody (hMAb) development.

Several hMAbs have been reported that have broad activity against a wide range of influenza viruses, including those that target the stem region of HA, which appear to be at least partially dependent on Fc-mediated effector functions for its in vivo activity [16,17,18,19], and which have driven major efforts in developing vaccines that promote the development of HA stem-targeting antibodies [20,21]. Additionally, in recent years, NA-specific hMAbs that have broad antiviral activity have also been described with variable dependence on Fc-mediated effector functions [22,23,24]. It is anticipated that NA-specific hMAbs that potently bind to the enzymatic site and inhibit the catalytic activity of NA should directly inhibit influenza viral infection, with minimal dependence on Fc-mediated effector activity. 

We have previously developed 1092D4, a fully human hMAb that is able to inhibit NA enzymatic activity and directly neutralize a wide range of IBV isolates, including both the Victoria and Yamagata lineages [25]. To determine if 1092D4 could directly mediate anti-viral activities in vivo, or if its in vivo activity is dependent on Fc-mediated effector functions, an engineered version of the hMAb was developed to reduce FcR binding. Specifically, it was produced with the combined L234A and L235A mutations in the lower hinge of Fc that form part of the FcγR binding site [26], and the P329G mutation that is part of the C1q binding site [27], referred to as LALAPG. Using the LALAPG approach, which diminishes FcR binding and complement binding and fixation [28,29,30,31,32,33], we were able to assess the dependence of 1092D4 on Fc-mediated effector activity in a lethal model of IBV infection and its activity against recent IBV isolates. 

## 2. Materials and Methods

Cells and viruses: Madin-Darby canine kidney (MDCK; IRR FR-926) and human embryonic kidney (HEK293T; ATCC CRL-11268) cells were grown in Dulbecco’s modified Eagle’s medium (DMEM; Corning, Mediatech, Inc. Durham, NC, USA) supplemented with 5% fetal bovine serum (FBS, VWR) and 1% PSG (penicillin, 100 units/mL; streptomycin, 100 μg/mL; L-glutamine, 2 mM, Corning, Mediatech, Inc.) at 37 °C with 5% CO_2_. For IBV infection, MDCK cells were maintained in post-infection medium (DMEM, 0.3% bovine albumin (BA), 1% PSG, and 1 μg/mL tosylsulfonyl phenylalanyl chloromethyl ketone (TPCK)-treated trypsin (Sigma, St. Louis, MO, USA)) and incubated at 33 °C with 5% CO_2._ Transfected HEK293T cells were maintained in DMEM with 5% Fetal Clone II (Cytiva, Marlborough, MA, USA) and 1× antibiotic/antimycotic (Corning, Mediatech) at 37 °C in a 5% CO_2_ incubator. Influenza B/Malaysia/2506/04, B/Florida/04/06, and B/New York/PV00081/18 were kindly provided from Dr. Florian Krammer at Icahn School of Medicine at Mount Sinai. B/Brisbane/60/2008 was previously described [34].

hMAb production. A new antibody vector was created by cutting out the IgG1 constant region of AbVec hIgG1 (GenBank: FJ475055.1) using SalI and HindIII restriction enzymes and replacing it with a newly synthesized oligonucleotide (IDT) containing the L234A, L235A, and P329G (LALAPG) mutations. Insertion was confirmed via sequencing. The previously cloned hMAb 1092D4 [25] underwent restriction digest with AgeI and SalI enzymes, was ligated into the new LALAPG vector, and was transformed into chemically competent DH5α *E. coli* cells. Resulting colonies were again sequenced to ensure the correct insertion. Purified plasmid DNA of either the parental heavy or the LALAPG heavy chain and parental light chains were transfected into 60–80% confluent HEK293T cells in 10 cm tissue culture treated dishes using JetPrime (Polyplus, Graffenstaden, France) as previously described [25]. Transfected cells were maintained for 8 days at 37 °C in a 5% CO_2_ atmosphere while culturing in DMEM with 5% Fetal Clone II (Cytiva) and 1× antibiotic/antimycotic (Corning/Mediatech). Media were harvested and replaced every two to three days. Harvested media were concentrated using 100,000 MWCO Amicon Ultra centrifugal filters (Millipore-Sigma, Burlington, MA, USA) and then purified using Protein A magnetic beads (Promega, Madison, WI, USA) as described in the product literature.

Enzyme-linked immunosorbent assay (ELISA): ELISA plates (Nunc MaxiSorp; Thermo Fisher Scientific, Grand Island, NY, USA) were coated with recombinant NA of B/Florida/04/2006 (BEI Resources, Manassas, VA, USA), B/Colorado/06/2017 (E Enzyme LLC, Gaithersburg, MD, USA), or B/Austria/1359417/2021 (Sino Biologicals, Wayne, PA, USA) at 2 µg/mL. hMAbs were diluted in PBS, and binding was detected with HRP-conjugated anti-human IgG (Jackson ImmunoResearch, West Grove, PA, USA).

FcγRI binding. Both 1092D4 and 1092D4–LALAPG were coated onto ELISA plates (Nunc MaxiSorp; Thermo Fisher Scientific, Grand Island, NY, USA) at either 2, 1, 0.5, or 0.25 µg/mL. Plates were sealed and stored at 4 °C overnight before being washed 3× in PBS. Biotinylated Human Fc gamma RI/CD64 (Acro Biosystems, Newark, DE, USA) was diluted to 1, 0.1, and 0 µg/mL, and 50 µL was added to wells in triplicate for each antibody concentration. After 1 h of shaking at room temperature, plates were washed 5× in PBS with 0.05% tween, and 50 µL of 2 µg/mL Streptavidin–horseradish peroxidase (Jackson ImmunoResearch, West Grove, PA, USA) was added and incubated for 1 h, shaking at room temperature. Plates were again washed 5× in PBS with 0.05% tween, and 50 µL of TMB substrate (SureBlue; KPL, SeraCare, Milford, MA) was added to wells and plates incubated for 5 min at room temperature. To stop the reaction, 50 µL of 1N HCl was added to wells and absorbance was read at 450 nm.

Antibody-dependent cellular phagocytosis (ADCP): ADCP activity of the hMAbs was measured as previously described [35,36] with slight modifications. Briefly, B/Colorado/06/2017 or B/Austria/1359417/2021 NA protein was biotinylated with the Biotin-XX Microscale Protein Labeling Kit (Life Technologies, Carlsbad, CA, USA). A total of 0.25 μg of biotinylated Ag or ~0.32 μg of BSA (used as a baseline control in an equivalent number of Ag molecules/bead) was incubated overnight at 4 °C with 1.9 × 10^6^ yellow-green neutravidin-fluorescent beads (Life Technologies) per reaction in a 25 μL of final volume. Antigen-coated beads were subsequently washed twice in PBS–BSA (0.1%) and transferred to a 5 mL Falcon round bottom tube (Thermo Fisher Scientific, Waltham, MA, USA). hMAbs, diluted at 0.5 or 5 μg/mL, were added to each tube in a 20 μL of reaction volume and incubated for 2 h at 37 °C in order to allow Ag–Ab binding. Then, 250,000 THP-1 cells (human monocytic cell line obtained from NIH AIDS Reagent Program) were added to the cells and incubated for 3 h at 37 °C. At the end of incubation, 100 μL 4% paraformaldehyde was added to fix the samples. Cells were then assayed for fluorescent bead uptake by flow cytometry using a BD Biosciences Symphony. The phagocytic score of each sample was calculated by multiplying the percentage of bead positive cells (frequency) by the degree of phagocytosis measured as mean fluorescence intensity (MFI) and divided by 10^6^. Values were normalized to background values (cells and beads without mAb) and an isotype control to ensure consistency in values obtained on different assays. Finally, the phagocytic score of the testing hMAb was expressed as the fold increase over BSA-coated beads.

Enzyme linked lectin assay (ELLA). ELLA were performed essentially as previously described by Bernard et al. [37]. In short, 96-well flat bottom immune plates (Nunc, ThermoFisher) were coated with 50 µg/mL Fetuin in PBS, sealed, and refrigerated overnight at 4 °C. The next day, plates were washed three times with PBS–Tween (0.05%). Assay optimization was performed using rNA of B/Colorado/06/2017 (E Enzyme LLC) by conducting 1:2 serial dilutions in DPBS (with calcium and magnesium) with 1% BSA to cover the range of 0.003906 to 8 µg/mL. Plates were sealed and incubated for 18 h at 37 °C before being washed 6 times with PBS–Tween. A total of 50 µL of peroxidase-conjugated lectin of *Arachis hypogaea* (peanut) was added to plates (2 µg/mL) and incubated for 2 h in the dark. Plates were washed three times, and 50 µL of TMB substrate (SureBlue; KPL, SeraCare) was added to wells and plates incubated for 20 min at room temperature. To stop the reaction, 50 µL of 1N HCl was added to wells and absorbance was read at 450 nm. A concentration of rNA was selected that was approximately 90% of the peak of the linear range of the resulting curve (~0.03 µg/mL) for inhibition assays. Both the 1092D4 parental and LALAPG variant hMAbs were diluted 1:2 on a washed fetuin-coated plate to cover a range from 0.006 to 12 µg/mL. The rNA was then added at the predetermined concentration to all but the “background” wells. Plates were sealed and incubated for approximately 20 h at 37 °C before being washed and developed as described above. Optical densities were corrected using the mean of the “background” wells, and then percent inhibition was calculated from the mean of wells incubated with the rNA alone. The IC_50_ was determined using non-linear regression in GraphPad Prism (Version 9.5.1).

Microneutralization assay. The neutralizing activity of both 1092D4 and 1092D4–LALAPG hMAbs against B/Malaysia/2506/04, B/Florida/04/06, B/Brisbane/60/2008, and B/New York/PV00081/18 was performed using virus neutralization assay as previously described [25]. Briefly, confluent monolayers of MDCK cells (5 × 10^4^ cells/well, 96-well plate format, quadruplicates) were infected with 200 PFU/well of the indicated virus. After 1 h of viral adsorption, cells were maintained in a post-infection medium (DMEM, 0.3% BA, 1% PSG, and 1 μg/mL tosylsulfonyl phenylalanyl chloromethyl ketone (TPCK)-treated trypsin (Sigma)) containing two-fold serial dilutions (starting concentration of 50 µg/mL) of the hMAbs and incubated at 33 °C with 5% CO_2_. At 72 h post-infection, virus neutralization titer (NT) was evaluated following crystal violet staining of the cells and expressed as the lowest concentration of the hMAb to prevent the virus-induced cytopathic effect (CPE). 

In vivo studies: The prophylactic activities of both 1092D4 and 1092D4–LALAPG hMAbs were tested against B/Malaysia/2506/04 in mice. Briefly, female C57BL/6 mice (8 weeks of age) were purchased from the Jackson Laboratory and maintained in the animal facility at Texas Biomedical Research Institute under specific pathogen-free conditions and ABSL2 containment. Mice were intraperitoneal treated with a single dose of 2 or 20 mg/kg of each hMAb, isotype control, or PBS (mock treated) 12 h prior to viral challenge. For virus infection, mice were anesthetized following gaseous sedation in an isoflurane chamber and inoculated intranasally with 10^4^ plaque-forming units (PFU)/mouse of B/Malaysia/2506/04. At days 2 and 4 post-infection (p.i.), viral replication was determined in the lungs of the infected mice. Briefly, four mice from each experimental group were euthanized by administration of a lethal dose of avertin and exsanguination, and lungs were surgically extracted and homogenized in 1 mL of PBS using Precellys tissue homogenizer (Bertin Instruments, Rockville, MD, USA). Virus titers (PFU/mL) were determined by standard plaque assay as previously described [25]. Geometric mean titers and data representation were performed using GraphPad Prism (v9.0). For the body weight and survival study, mice (N = 5, group) were monitored daily for 10 days p.i. for morbidity (body weight) and mortality (survival rate). Mice showing a loss of more than 25% of their initial body weight were defined as reaching the experimental end point and humanely euthanized. All animal protocols were approved by Texas Biomedical Research Institute (IACUC 1785 MU 0). 

## 3. Results

### 3.1. 1092D4–LALAPG Has Comparable Binding to IBV NA

To minimize the ability of 1092D4 IgG1 to bind to FcγRs, the well-described L234A and L235A substitutions (LALA) to residues in the CH2 domain, which form part of the FcγR binding site, and the P329A substitution (PG) in the C1q binding site to minimize binding to C1q [26,33,38,39] were made into the heavy chain of 1092D4, resulting in 1092D4–LALAPG. 1092D4–LALAPG retained comparable binding to IBV NA from B/Florida/04/2006 (Yamagata), B/Colorado/06/2017 (Victoria), and B/Austria/1359417/2021 (Victoria) rNA proteins, with no significant differences apparent (Figure 1).

### 3.2. 1092D4–LALAPG Reduces FcγRI Binding and Antibody-Dependent Cellular Phagocytosis

The LALAPG mutation is expected to reduce the ability of 1092D4 to bind FcR and subsequently diminish Fc-mediated effector functions. As expected, 1092D4-LALPG exhibited minimal binding to the FcγRI as compared to 1092D4 (Figure 2A). Similarly, while 1092D4 exhibited antibody-dependent cellular phagocytosis (ADCP) activity of NA B/Colorado/06/2017 and NA B/Austria/1359417/2021 coated beads, 1092D4–LALAPG had minimal ADCP activity (Figure 2B). These results confirm that the FcR binding and subsequent ability to mediate Fc-effector functions of 1092D4–LALAPG are greatly diminished. 

### 3.3. 1092D4–LALAPG Retains Ability to Inhibit NA Enzymatic Activity

The primary function of influenza NA is to cleave sialic acid residues from the cell surface, enabling the release of the mature virus which can then infect new cells. Consistent with our previous findings that 1092D4 can potently inhibit NA enzymatic activity [25], 1092D4–LALAPG inhibited the activity of NA in the enzyme-linked lectin assay (ELLA) (Figure 3). Both 1092D4 and 1092D4–LALAPG potently inhibited the activity of rNA from B/Colorado/06/2017, with comparable IC_50_ of 0.049 and 0.037 μg/mL, respectively. These results indicate that the ability to inhibit the enzymatic activity of NA is not Fc-dependent. 

### 3.4. 1092D4 and 1092D4–LALAPG Potently Neutralize Diverse IBV

The ability of 1092D4 and 1092D4–LALAPG to inhibit viral infection was measured using a microneutralization assay. Both were able to similarly neutralize all IBVs tested at NT < 1.0 μg/mL, including the Victoria lineage strains B/Malaysia/2506/2004, B/Brisbane/60/2008, and the recent B/New York/PV00081/2018; and the Yamagata lineage strain B/Florida/04/2006 (Table 1). NTs for the viruses were comparable between 1092D4 and 1092D4–LALAPG, with the exception of B/Brisbane/60/2008, for which 1092D4–LALAPG had a slightly higher NT (0.15 vs. 0.039 μg/mL). These results indicate that neutralization of IBVs by 1092D4 is not Fc-dependent.

### 3.5. 1092D4–LALAPG Protects from Lethal IBV Infection

Considering that the potent in vitro activity of 1092D4 is not compromised when expressed as 1092D4–LALAPG, we sought to determine if its in vivo efficacy was dependent on Fc-effector mechanisms. Mice received a single intraperitoneal injection of 1092D4 or 1092D4–LALAPG 12 h prior to challenge with a lethal dose of B/Malaysia/2506/04. All IBV-infected control mice (isotype control and mock-treated groups) required euthanasia by 7 days p.i., while all 1092D4- and 1092D4–LALAPG-treated mice survived infection (Figure 4A). All IBV-infected mice that were treated with isotype control hMAb or mock-treated had progressive weight loss that reached the 75% endpoint threshold by 7 days p.i. (Figure 4B). Mice treated with 1092D4 (2 and 20 mg/kg) or 20 mg/kg of 1092D4–LALAPG maintained their bodyweight, while those mice treated with low-dose 1092D4–LALAPG exhibited modest, transient ~10% body weight loss compared to mice treated with 2 mg/kg 1092D4, which did reach statistical significance (*p* = 0.0135). All doses of 1092D4 and 1092D4–LALAPG resulted in significant reduction in lung virus titers at 2 dpi, with no significant difference between equal doses of 1092D4 and 1092D4–LALAPG (Figure 4C). At 4 dpi, 1092D4 (2 and 20 mg/kg) and 20 mg/kg 1092D4–LALAPG had significantly lower viral titer compared to isotype control treated mice, with no significant difference between equal doses of 1092D4 and 1092D4–LALAPG. Together, these results indicate that 1092D4 and 1092D4–LALAPG provide similar protection from lethal IBV infection, suggesting 1092D4 can mediate effective protection against IBV without dependence on Fc-effector functions. 

## 4. Discussion

Improved vaccines and therapeutics for influenza that are sufficiently robust to tolerate its continued viral mutations and antigenic drift variation are needed. The pre-clinical and clinical efficacy of NA-directed antibodies and antivirals highlight the therapeutic potential of targeting NA. Here, we demonstrate that a hMAb that has activity against a broad range of IBV isolates potently inhibits NA enzymatic activity and can mediate protection against lethal IBV challenge without the reliance on Fc-effector functions.

The ability of Abs to mediate anti-viral activity through Fc-dependent activities such as antibody-dependent cellular cytotoxicity (ADCC), antibody-dependent cellular phagocytosis (ADCP), and complement-mediated processes, as well as Fc-independent activities such as neutralization and viral enzyme inhibition, provides multiple and often complementary mechanisms by which the humoral immunity can respond to viruses. Fc-dependent effector functions are particularly critical for the anti-viral activity of Abs that do not recognize epitopes that can result in direct neutralization or inhibition of virus activity. A concern for such Fc-dependent Abs is extensive neutrophil, eosinophil, and NK cell accumulation in the lungs along with alveolar macrophage activation, which could result in hyperinflammation and the associated cytokine storm and pathology that negates the antiviral activity of the Ab [40,41,42,43]. Additional considerations for reliance on Fc-mediated anti-viral Ab activity for protection include an over-dependence on multiple components of the immune response which might be diminished in some individuals (elderly and immunocompromised), and the potential for antibody-dependent enhancement (ADE) of infection, which has been described previously for influenza [43,44,45,46]. For vaccine and anti-viral therapeutic mAb development, Abs that have direct anti-viral activity are the most desirable. 

A few highly conserved epitopes on the influenza virus have been identified which have garnered attention with respect to universal vaccine development, including the stem region of HA and the enzymatic active site of NA. Thus far, HA stem Abs predominantly require Fc-effector functions for in vivo activity [17,18,47,48], while the dependence on Fc-effector functions of NA active site-specific Abs is variable and likely determined by the potency of direct NA-inhibitory activity. Our results clearly demonstrate that 1092D4 hMAb, which is able to potently inhibit NA activity, is protective against lethal IBV infection even when its ability to engage FcR has been greatly diminished. This is consistent with previous results for polyclonal and monoclonal N1 Abs with NA-inhibitory activity that could protect from H1N1 infection in the absence of FcR engagement; however, an N1 mAb that lacks NA-inhibitory activity could not [22,49]. Similar results were reported for N9 mAbs, with mAbs with weak NA inhibition having greater dependence on Fc-effector function than those with stronger NA inhibition [24]. The potential of NA Ab response to protect without dependence on Fc-effector mechanisms has been further extended to immunization with NA [50]. NA Abs which are highly neutralizing/inhibiting are important in reducing influenza outbreaks. Tan et al, [51] demonstrated that guinea pigs infected with B/Malaysia/2506/2004, when administered the anti-NA IBV murine Ab 1F2, had much lower transmission to close-contact cage inhabitants than untreated animals (40% vs. 100% transmission). This reduction occurred even without impacting viral replication or the duration of viral shedding, highlighting the need to further investigate the in vivo antiviral mechanisms of NA antibodies. The NA-inhibitory activity of 1092D4 suggest that it recognizes an epitope within or near the active site, but structural analysis beyond the scope of this paper is needed to definitively resolve this.

Reducing the Fc-effector function of the previously identified NA-specific hMAb 1092D4 did not impact its capacity to bind or inhibit NA activity, nor did it alter the ability to neutralize influenza B in vitro. This lack of difference also persisted when antibodies were administered to mice prior to infection with a lethal dose of B/Malaysia/2506/04. Although Fc-effector function of 1092D4 can likely contribute to some of the viral clearance, we can conclude from this work that direct inhibition of NA through binding is likely a major anti-viral mechanism of 1092D4. 

NA-inhibiting antibodies may be key prophylactic and/or therapeutic agents, especially in the young or immunocompromised individuals, where other therapies are not available or are ineffective as a result of emerging drug-resistant viral variants. Research on such antibodies deserves further consideration. 

## Figures and Tables

**Figure 1 viruses-15-01540-f001:**
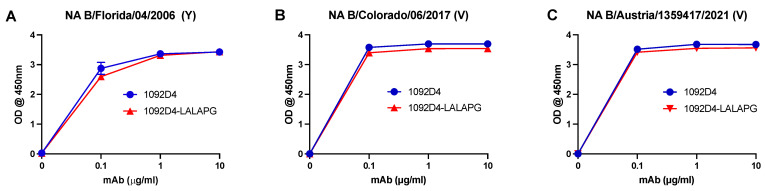
The LALAPG mutation in the Fc does not impact binding in IBV NA proteins. Increasing concentrations of the hMAbs 1092D4 and 1092D4 with the LALAPG mutation were tested in triplicate for binding to the NA proteins of (**A**) B/Florida/04/2006 (Yamagata), (**B**) B/Colorado/06/2017 (Victoria), and (**C**) B/Austria/1359417/2021 (Victoria) by ELISA.

**Figure 2 viruses-15-01540-f002:**
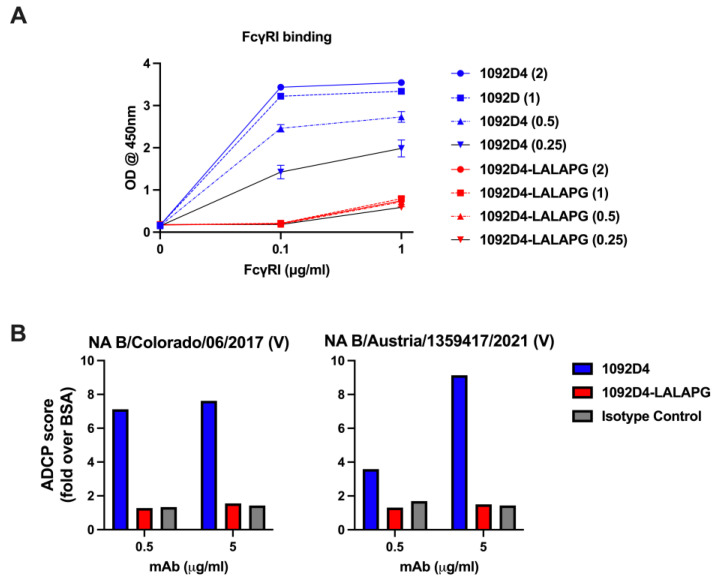
The LALAPG mutation reduces FcγRI binding and antibody-dependent cellular phagocytosis. (**A**) Increasing concentrations of the hMAbs 1092D4 and 1092D4–LALAPG (μg/mL) were tested in triplicate for binding to increasing concentrations of FcγRI by ELISA. (**B**) Ab-dependent cellular phagocytosis (ADCP) assay. NA-coated and BSA-coated beads were incubated with hMAb and then added to THP-1 cells. After incubation, cells were assayed for fluorescent bead uptake by flow cytometry. The ADCP score of each hMAb was calculated by multiplying the percentage of bead-positive cells (frequency of phagocytosis) by the mean fluorescence intensity (MFI) of the beads (degree of phagocytosis) and dividing by 10^6^.

**Figure 3 viruses-15-01540-f003:**
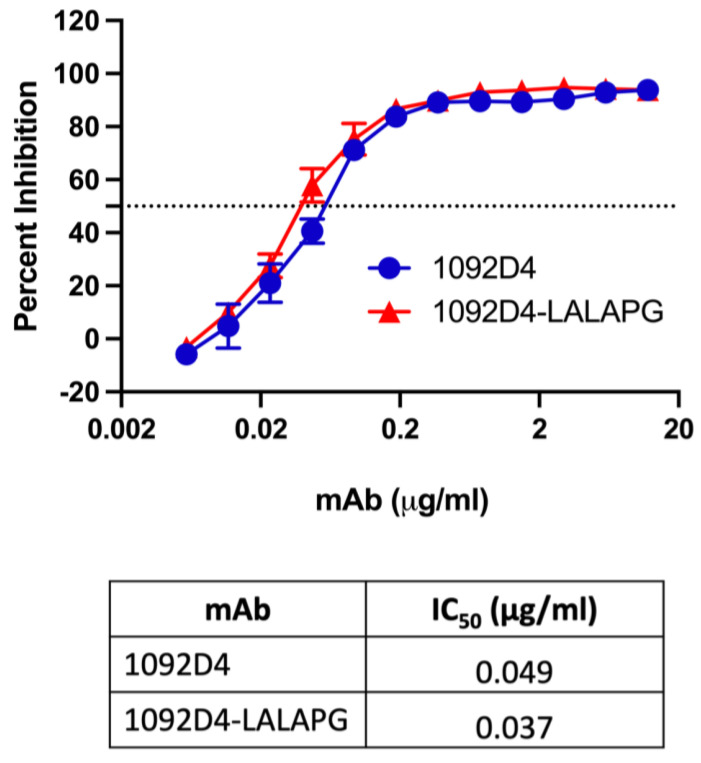
Similar NA inhibition between the 1092D4 and the LALAPG expressing Fc mutant. Inhibition of rNA (B/Colorado/06/2017) activity was compared between 1092D4 and 1092D4 with the LALAPG mutation. The rNA was added with two-fold serial dilutions of the hMAbs, in duplicate, on fetuin-coated plates, and NA activity was determined at 18 h using peroxidase conjugated lectin of *Arachis hypogaea* and TMB substrate. Data represent mean percentages of NA-alone activity from duplicate wells.

**Figure 4 viruses-15-01540-f004:**
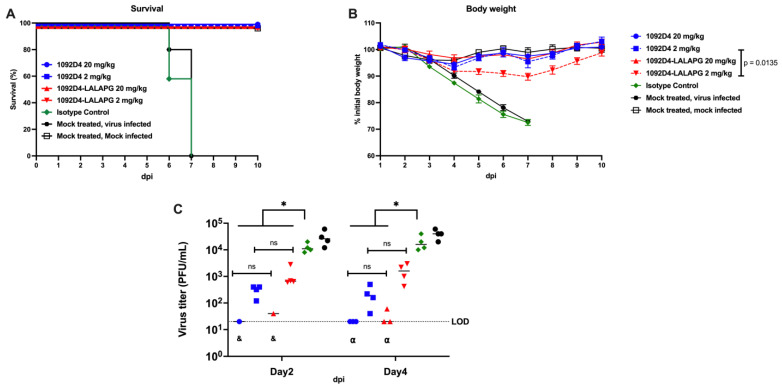
In vivo prophylactic activity of the hMAbs 1092D4 and 1092D4 with the LALAPG mutation. C57Bl/6 mice (8-weeks-old) were given either 20 or 2 mg/kg of indicated hMAbs or PBS (mock-treated), intraperitoneally, 12 h before infection. Mice were then challenged with 10^4^ PFU of B/Malaysia/2506/04 and monitored daily for 10 days p.i. for survival (**A**) and body weight loss (**B**) (N = 5, group). Significance between equal mAb doses were determined via two-way ANOVA. To evaluate viral replication in the lungs (**C**), mice were sacrificed at 2 (N = 4, group) and 4 (N = 4, group) days p.i., and whole lungs were used to quantify viral titers by standard plaque assay (PFU/mL). Each symbol represents an individual mouse. * indicates significant differences (*p* < 0.05 using one-way ANOVA and Tukey’s test). "ns” indicates non-significant statistical difference (*p* > 0.05). Dotted line indicates the limit of detection (LOD) of the assay (20 PFU/mL). The “&” symbol indicates that the virus was not detected in three mice per group. The “α” symbol indicates that the virus was not detected in only one mouse per group.

**Table 1 viruses-15-01540-t001:** Ability of the hMAbs 1092D4 and 1092D4 with the LALAPG mutation to inhibit viral infection. Microneutralization assay. MDCK cells were infected with the indicated virus and then incubated with two-fold serial dilutions (starting concentration, 50 µg/mL) of the hMAbs. Virus NT was evaluated 72 h post infection following crystal violet staining and expressed as the lowest concentration of the hMAb to prevent virus-induced CPE. Mock-infected cells and viruses in the absence of hMAb were used as internal controls.

	Neutralization (NT_50_ μg/mL)	
Virus	1092D4	1092D4–LALAPG
B/Malaysia/2506/04	0.39	0.39
B/Florida/04/06	0.39	0.39
B/Brisbane/60/2008	0.04	0.15
B/New York/PV00081/18	0.63	0.63

## Data Availability

The authors confirm that the data supporting the findings of this study are available within the article.

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
