# Peer review of "Fc-Effector-Independent in vivo Activity of a Potent Influenza B Neuraminidase Broadly Neutralizing Antibody"

_viruses, 2023, doi:10.3390/v15071540_

Round 1

Reviewer 1 Report

This study seeks to determine if the activity of an anti-influenza B virus (IBV) neuraminidase (NA) antibody is dependent on the Fc effector region. They introduced LALA-PG mutations and measured binding, inhibition of NA activity, neutralization and efficacy as a prophylactic against lethal IBV challenge in mice. They found that LALA-PG did not impact these aspects of the antibody. Overall this study is worthwhile but I feel like it is better suited to a short communication format rather than a research article.

Comments as follows;

Abstract – no mention of polymerase inhibitors?

Line 21 – limiting death was not discerned in this study as you designated humane endpoints

Line 55-56 – vaccines twice

Line 63 – suggest describing LALA-PG a bit more in the introduction for those outside the field.

Suggest a very clear statement as to the goal or aims of the study for ease of reading in the introduction. I feel the goal of the study could be made clearer. 

If the goal of the study is to determine if the efficacy of 1092D4 was dependent on Fc-effector function, it should be made clear as to the significance of why this is important in the abstract and the introduction.

From your previous study (25), which describes a number of antibodies, why did you choose to focus on 1094D4 here and not the others? Both 1092D4 and 1086F8 were quoted to have potential for further evaluation.

In the in vitro studies section of materials and methods, it’s stated that virus titers in PFU/mL were determined (line 150). It should be stated the total volume of the lung homogenates used and if the lungs were weighed.

Figure 3B. It’s not clear what the p-value stated in the legend is referring to. Suggest to be more specific.

Figure 3C. It’s not clear why there are data points at 0 and at the LOD. There are no asterisks on this figure, although they are alluded to in the figure legend. Further, is the LOD 20PFU/mL or 20PFU?

Discussion on the putative locations of binding of these antibodies would be informative and add to the paper.

Minor editing of English language required

Author Response

Reviewer #1

This study seeks to determine if the activity of an anti-influenza B virus (IBV) neuraminidase (NA) antibody is dependent on the Fc effector region. They introduced LALA-PG mutations and measured binding, inhibition of NA activity, neutralization and efficacy as a prophylactic against lethal IBV challenge in mice. They found that LALA-PG did not impact these aspects of the antibody. Overall this study is worthwhile but I feel like it is better suited to a short communication format rather than a research article.

  1. Abstract – no mention of polymerase inhibitors?

Response: As the relative effectiveness of polymerase inhibitors against IBV is still not thoroughly settled, we have opted to discuss this further in the introduction instead of briefly mentioning it in the Abstract.

  1. Line 21 – limiting death was not discerned in this study as you designated humane endpoints

Response: This line has been changed to “…protection from lethal challenge.”, to more accurately reflect the manner in which the experiment was performed.

  1. Line 55-56 – vaccines twice

Response: This typo has been corrected in the revised manuscript.

  1. Line 63 – suggest describing LALA-PG a bit more in the introduction for those outside the field.

Response: Additional details describing LALAPG have been added to the final paragraph of the introduction in the revised manuscript.

  1. Suggest a very clear statement as to the goal or aims of the study for ease of reading in the introduction. I feel the goal of the study could be made clearer.

Response: The final paragraph of the introduction has been revised to clarify the goal of the study.

  1. If the goal of the study is to determine if the efficacy of 1092D4 was dependent on Fc-effector function, it should be made clear as to the significance of why this is important in the abstract and the introduction.

Response: We have provided additional clarification in the Abstract and Introduction as to the significance of evaluated Fc-effector function dependence.

  1. From your previous study (25), which describes a number of antibodies, why did you choose to focus on 1094D4 here and not the others? Both 1092D4 and 1086F8 were quoted to have potential for further evaluation.

Response: In our previous study (25, PMC6414695) 1092D4 exhibits slightly better in vitro and in vivo activity as compared to 1086F8, which was our basis for selecting it for the more in-depth characterization now included in this revised manuscript.

  1. In the in vitro studies section of materials and methods, it’s stated that virus titers in PFU/mL were determined (line 150). It should be stated the total volume of the lung homogenates used and if the lungs were weighed.

Response: The volume of PBS used for lung homogenization was mentioned in the revised manuscript (line 204).

  1. Figure 3B. It’s not clear what the p-value stated in the legend is referring to. Suggest to be more specific.

Response: Figure 3 legend has been revised to clarify which comparisons the stated p-values refer to.

  1. Figure 3C. It’s not clear why there are data points at 0 and at the LOD. There are no asterisks on this figure, although they are alluded to in the figure legend. Further, is the LOD 20PFU/mL or 20PFU?

Response: The figure was revised to clearly indicate the undetected points, those that were detected at the LOD (20 PFU/ml), and the asterisk. All changes were included in the revised manuscript (Figure 4C, lines 304-310). The asterisks were lost upon conversion to PDF and are now apparent in revised manuscript.

  1. Discussion on the putative locations of binding of these antibodies would be informative and add to the paper.

Response: We have included additional discussion of potential of 1092D4 epitope to reside within or near the active of site of NA, and indicated that structural analysis beyond the scope of this paper would be need to definitively resolve the epitope.

Reviewer 2 Report

Comments:

In this manuscript, to discern if the in vivo efficacy of 1092D4 was dependent on Fc-effector function, 1092D4 hMAb with reduced ability to bind to Fc receptors (1092D4-LALAPG) was generated and tested. It is found that the 1092D4-LALAPG had comparable in vitro binding, neutralization, and inhibition of NA activity to 1092D4. 1092D4-LALAPG was effective at limiting weight loss, viral burden, and death from a lethal challenge of IBV in mice.

Overall, the manuscript is well-organized and follows a logical structure that is easy to follow. The authors presented the story in a clear and concise manner.

1.     The Fc-receptors are double blade sword, antibodies could mediate anti-viral activity through Fc-receptors such as antibody-dependent cellular cytotoxicity (ADCC), antibody dependent cellular phagocytosis (ADCP), and complement mediated processes. A concern for such Fc-dependent Abs is extensive neutrophil, eosinophil, and NK cell accumulation in the lungs along with alveolar macrophage activation which could result in hyperinflammation and the associated cytokine storm and pathology that negates the anti-viral activity of the Ab. Therefore, it is important to investigate the mechanism of action for the 1092D4 hMAb, and 1092D4-LALAPG as reduced ability to bind to Fc receptors was generated. It is necessary to show 1092D4-LALAPG has reduced ability to bind to Fc receptors as compared to 1092D4 hMAb. 

2.     Another interesting thing is that is there any effectors been activated by the 1092D4 hMAb? If there is any, what are they? And how they have been activated? And what are the consequences?

Author Response

Reviewer #2

  1. The Fc-receptors are double blade sword, antibodies could mediate anti-viral activity through Fc-receptors such as antibody-dependent cellular cytotoxicity (ADCC), antibody dependent cellular phagocytosis (ADCP), and complement mediated processes. A concern for such Fc-dependent Abs is extensive neutrophil, eosinophil, and NK cell accumulation in the lungs along with alveolar macrophage activation which could result in hyperinflammation and the associated cytokine storm and pathology that negates the anti-viral activity of the Ab. Therefore, it is important to investigate the mechanism of action for the 1092D4 hMAb, and 1092D4-LALAPG as reduced ability to bind to Fc receptors was generated. It is necessary to show 1092D4-LALAPG has reduced ability to bind to Fc receptors as compared to 1092D4 hMAb.

Response:  We appreciate the reviewer’s very thoughtful and helpful comments. We have subsequently demonstrated the reduced binding of 1092D4-LALAPG to FcγRI, and this is now presented as Figure 2A.

  1. Another interesting thing is that is there any effectors been activated by the 1092D4 hMAb? If there is any, what are they? And how they have been activated? And what are the consequences?

Response: We have demonstrated in this paper, and in our previous paper on 1092D4 that is able to potently and directly neutralize IBV, and with this paper demonstrate the reduction in Fc-effector mediated functional activity does not significantly impact its in vivo activity. We had not previously specifically characterized the Fc-effector function of 1092D4, and in response to the reviewer’s suggestion now demonstrate that 1092D4 can mediate antibody-dependent cellular phagocytosis (ADCP) by the THP-1 monocyte cell line (Figure 2B), representing a prototypical Fc effector function. And as anticipated, the ADCP activity of 1092D4-LALAPG is minimal.